# Comparative analyses of adsorbed circulating proteins in the PMMA and PES hemodiafilters in patients on predilution online hemodiafiltration

Md. Shoriful Islam[1,2,3☯], Shingo Ema[4☯], Md. Mahamodun Nabi[1], Md. Muedur Rahman[1,3], A. S. M. Waliullah[1], Jing Yan[1], Rafia Ferdous[1], Takumi Sakamoto[1,3], Yutaka Takahashi[1,3], Akihiko Kato[5], Tomohito Sato[1,6], Tomoaki Kahyo[1,7], Mitsutoshi Setou[1,7]*

1 Department of Cellular and Molecular Anatomy, Hamamatsu University School of Medicine, Hamamatsu, Shizuoka, Japan, 2 Department of Pharmacy, Islamic University, Kushtia, Bangladesh, 3 Preppers Co. Ltd., Hamamatsu University School of Medicine, Hamamatsu, Shizuoka, Japan, 4 Medical Device Management Department, Hamamatsu University Hospital, Hamamatsu, Shizuoka, Japan, 5 Blood Purification Unit, Hamamatsu University Hospital, Hamamatsu, Shizuoka, Japan, 6 First Department of Surgery, Hamamatsu University School of Medicine, Hamamatsu, Shizuoka, Japan, 7 International Mass Imaging and Spatial Omics Center, Institute of Photonics Medicine, Hamamatsu University School of Medicine, Hamamatsu, Shizuoka, Japan

☯ These authors contributed equally to this work.

* setou@hama-med.ac.jp

**Data Availability Statement:** All relevant data are within the manuscript and its Supporting Information files.

## Abstract

Acute and chronic inflammation are common in patients with end-stage kidney disease (ESKD). So, the adsorption of pro-inflammatory cytokines by the hollow fiber of the dialysis membrane has been expected to modify the inflammatory dysregulation in ESKD patients. However, it remains to be determined in detail what molecules of fiber materials can preferably adsorb proteins from the circulating circuit. We aimed this study to analyze directly the adsorbed proteins in the polymethyl methacrylate (PMMA) and polyethersulfone (PES) membranes in patients on predilution online hemodiafiltration (OL-HDF). To compare the adsorbed proteins in the PMMA and PES hemodiafilters membrane, we initially performed predilution OL-HDF using the PES (MFX-25Seco) membrane while then switched to the PMMA (PMF™-A) membrane under the same condition in three patients. We extracted proteins from the collected hemodiafilters by extraction, then SDS-PAGE of the extracted sample, protein isolation, in-gel tryptic digestion, and nano-LC MS/MS analyses. The concentrations of adsorbed proteins from the PMMA and PES membrane extracts were 35.6±7.9 µg/µL and 26.1±9.2 µg/µL. SDS-PAGE analysis revealed distinct variations of adsorbed proteins mainly in the molecular weight between 10 to 25 kDa. By tryptic gel digestion and mass spectrometric analysis, the PMMA membrane exhibited higher adsorptions of β2 microglobulin, dermcidin, retinol-binding protein-4, and lambda-1 light chain than those from the PES membrane. In contrast, amyloid A-1 protein was adsorbed more potently in the PES membrane. Western blot analyses revealed that the PMMA membrane adsorbed interleukin-6 (IL-6) approximately 5 to 118 times compared to the PES membrane. These

**Funding:** This work was supported by the MEXT Project to promote public utilization of advanced research infrastructure (Imaging Platform) (Grant Number JPMXS0410300220) and by the AMED project (Grant Number 21ak0101179), Japan. The funders had no role in the study design, data collection and analysis, preparation of the manuscript, or decision to publish. There was no additional external funding received for this study.

**Competing interests:** The authors have declared that no competing interests exist.

findings suggest that PMMA-based OL-HDF therapy may be useful in controlling inflammatory status in ESKD patients.

## Introduction

The polymethyl methacrylate (PMMA) membrane has several beneficial features, including good biocompatibility [1], protein adsorption properties, and a uniform structure with homogenized pores [2, 3]. PMMA dialyzer membrane also provides an anti-inflammatory effect [4] and results in the improvement of renal anemia [5], itching [6], malnutrition [7], and poor immune response [8]. In addition, a nationwide cohort study in Japan demonstrated that regular hemodialysis (HD) with PMMA membrane dialyzer was significantly associated with a lower rate of 1- and 2-year mortality than that with polysulfone (PS) membrane [9]. PMMA membrane [10]. PMMA membrane was effective in mitigating pruritus in HD patients [11].

Dialysis-related inflammation is closely associated with poor outcomes in HD patients. Excessive generation of inflammatory mediators is thought to be implicated in disease progression in acute kidney damage. IL-6 plasma concentrations are frequently utilized to characterize acute inflammatory reactions caused by trauma or infection, including severe COVID-19 [12], especially in the early diagnosis of neonatal sepsis [13]. PMMA is a novel membrane material that claims to adsorb a variety of inflammatory cytokines. Adsorption and removal of inflammatory cytokines and other proteins by a PMMA membrane hemofilter improves the pathological status of the patients [14].

Recently, with technological advances [15, 16], analyses of plasma proteins have attracted increased interest. Surface-enhanced laser desorption/ionization time-of-flight mass spectrometry (SELDI-TOF-MS) demonstrated that the serum protein profile was quite different between HD patients and control subjects [17]. An *in vitro* experiment demonstrated that the PMMA membrane can adsorb pro-inflammatory cytokines more penitently than the PS membrane. In addition, the PMMA membrane adsorbs mainly in the molecular weight (MW) of cytokines from 10 to 30 kDa [18]. However, there was little study to examine the characteristics of adsorbed proteins in the dialyzer filter using mass spectrometry in regular dialysis patients.

In this study, we measured adsorbed proteins in the PMMA membrane hemodiafilter (Filtryzer® PMF™-A) and PES membrane (MFX-25S eco) by nano-HPLC MS/MS analyses in three patients on predilution online hemodiafiltration (OL-HDF), and compared the proteomics of absorbed proteins.

## Material and methods

### Reagents and chemicals

PMF and MFX membranes were purchased from Toray Medical Co. Ltd. and Nipro Co. Ltd., Japan, respectively. SDS, tris(hydroxymethyl) amino methane, Thiourea (M8T2663), and Tetramethylethylenediarnine (M6E6737) were purchased from Nacalai Tesque Inc. (Kyoto, Japan). Urea, Polyoxyethylene sorbitan monolaurate, LC-MS grade ultrapure water (CAS 7732-18-5), chloroform (CAS 67-66-3), methanol (CAS 67-56-1), Ethanol (CAS 64-19-7) Acetone, LC-MS grade ultrapure water (CAS 7732-18-5), and sample buffer solution (2ME+) were purchased from FUJIFILM Wako pure chemical industries (Osaka, Japan). Iodoacetamide (L11794) was purchased from KANTO CHEMICAL CO., INC. DTT from ALEXIS biochemicals. Trypsin gold, mass spectrometry grade (0000540881), was purchased from Promega. IL-6 polyclonal antibody (Cat 21865-1-AP) was purchased from Proteintech.

## Demographic and biochemical data

We enrolled three patients undergoing predilution OL-HDF thrice a week in this study. The recruitment period for this study was from February 28, 2022, to March 25, 2022. This study was approved by the Hamamatsu University School of Medicine's ethics committee (Approval No: 19–169). It was carried out following ethical standards and the spirit of the Declaration of Helsinki. Patients scheduled for hemodiafiltration provided written informed consent to use their used dialyzers in this study. The demographic and biochemical data information of the participating patients is summarized in Table 1.

## Collection of hemodiafilter

We collected used hemodiafilters just after the end of the dialysis session. The characteristics of each membrane are shown in Table 2. All patients underwent OL-HDF using either the PES or PMMA brand-new without any re-use filters at about 2-day intervals on a distinct day. After the OL-HDF session, we washed the blood and dialysate sides with 500 mL of normal saline and 500 mL of distilled water, respectively, and stored them at -80°C until the measurement.

**Table 1. Demographic and biochemical information of the three patients.**

| | Patient 1 | Patient 2 | Patient 3 |
|---|---|---|---|
| **Patient demographics** | | | |
| Age (year) | 50 | 70 | 68 |
| Gender | Female | Male | Female |
| Applied Membranes | PES first, then PMMA the following week. | PES first, then PMMA the next week. | PES first, then PMMA the following week. |
| BMI (kg/m$^2$) | 20.8 | 23.4 | 21.0 |
| Disease history | Chronic dialysis, SLE, AAA, Angina, MI, and malignant melanoma. | Acute dialysis, DM 2 nephropathy, Angina, and third-degree burns on both lower extremities. | Chronic dialysis, AH, HD, CRF, CA, Colon polyps, Inpatient of TAVI surgery at severe AS. |
| **Blood chemistry pre-dialysis** | | | |
| C-reactive protein (CRP) (mg/dl) | 0.41 | 2.15 | 15.67 |
| Creatinine (mg/dl) | 7.37 | 7.88 | 7.77 |
| MCV (fL) | 109 | 95 | 100 |
| MCH (pg) | 34.9 | 30.7 | 30.5 |
| MCHC (mg/dl) | 32.1 | 32.5 | 30.6 |
| Blood urea nitrogen (mg/dl) | 47.8 | 39.7 | 51.2 |
| ALP (U/L) | 130 | 113 | 97 |
| Uric Acid (mg/dl) | 6.7 | 7.2 | 6.7 |
| RBC (x10^4/μl) | 335 | 257 | 302 |
| WBC (/ μl) | 4690 | 6800 | 6970 |
| Na (mmol/L) | 141 | 134 | 136 |
| K (mmol/L) | 4.4 | 4.0 | 3.9 |
| Cl (mmol/L) | 107 | 100 | 99 |
| Ca (mg/dl) | 8.4 | 8.0 | 8.7 |

Biochemical information is taken before the dialysis of the patients with written consent. Inflammatory biomarker blood CRP level is higher for all the patients. Here, SLE = Systemic lupus erythematosus, AAA = Abdominal aortic aneurysm, AH = Alcoholic hepatitis, HD = Hashimoto's disease, CRF = Chronic renal failure, CA = Cerebral artery aneurysm (postoperative).

**Table 2. Characteristics of the PMMA & PES dialyzer membranes.**

| Characteristics | MFX-25 U eco | PMF-21A |
|---|---|---|
| Membrane material | PES | PMMA |
| Membrane bore (μm) | 200 | 200 |
| Membrane thickness (μm) | 40 | 30 |
| Effective membrane area (m$^2$) | 2.5 | 2.1 |
| UFR (mL/mmHg/hr.) | 91 | 56 |

Here, PES = Polyethersulfone, UFR = Ultrafiltration coefficient

## Extraction of hemodiafilter

Each hemodiafilter was broken down mechanically, and then the hollow fiber was cut into small pieces to perform extraction and dried by vacuum evaporation. A portion of 5 gm of each hemodiafilter was weighed into a 50 mL polypropylene tube. Then, 40 mL of strong chaotropic solubilization buffer (6M urea, 2M thiourea, 0.4% SDS, and one mM dithiothreitol) was added and incubated at room temperature for 1 hour. The samples were centrifuged for 10 minutes at 4°C at 15,000 rpm. The protein solution (supernatant of each dialyzer membrane extract) was precipitated by adding seven volumes of precipitation solution containing 50% ethanol, 25% methanol, and 25% acetone and then incubated at -80°C overnight. Upon centrifugation and washing, the protein pellet was reconstituted with 200 μl of a denaturing buffer (6 M urea in 100 mM Tris-HCl, pH 7.8). The resulting solution was vortexed and stored for the downstream application.

## Total protein concentration and SDS-PAGE

Total protein concentrations were assessed from all dialyzer membrane extracts using the Pierce$^{TM}$ BCA protein assay kit (Thermo Fisher Scientific, Waltham, MA, USA). The absorbance was taken at 562 nm on a microplate reader, and the protein concentration of each sample was calculated using the standard curve. The SDS–PAGE of all dialyzer membrane extracts was performed according to the method described by Laemmli [19] using a Bio-Rad system (Bio-Rad Laboratories, CA, USA). 20 μg samples were separated by SDS-PAGE on 14% acrylamide gel and then stained with Coomassie blue. The three gel pieces containing bands specific to the PMMA membrane were cut out on a clean bench (Lane 2, Spot 8, 10, and 12; Fig 1B and S1 Raw images). In addition, the gel pieces from the PES sample at the same size position (Lane 2, band 7, 9, and 11; Fig 1B and S1 Raw images) were also cut out and used as a control. Moreover, the twelve gel pieces containing bands specific for both membranes at the same size position were also cut out on a clean bench (Lane 1, band 1 to 6; Lane 3, band 13 to 18; Fig 1B and S1 Raw images). The cut-out gel pieces were placed in a 1.5 ml tube and stored in a -80° C freezer until in-gel protein digestion.

## In-gel tryptic protein digestion and peptide purification

In-gel protein digestion was performed according to the method described by Zheng YZ et al. [20] and 100 μl of reduction buffer (15 mM DTT, 25 mM ammonium bicarbonate) was added to an approximately 6 x 2 mm piece of acrylamide gel, shaken at 56°C for 60 min, and the solution was discarded. The gel was washed with 100 μl of wash buffer (25 mM ammonium bicarbonate) by shaking for 10 min at room temperature. Then, 100 μl of alkylation buffer (55 mM iodoacetamide, 25 mM ammonium bicarbonate) was added and shaken for 45 min at room temperature in the dark. The gel was washed once and twice with 100 μL of wash buffer (25

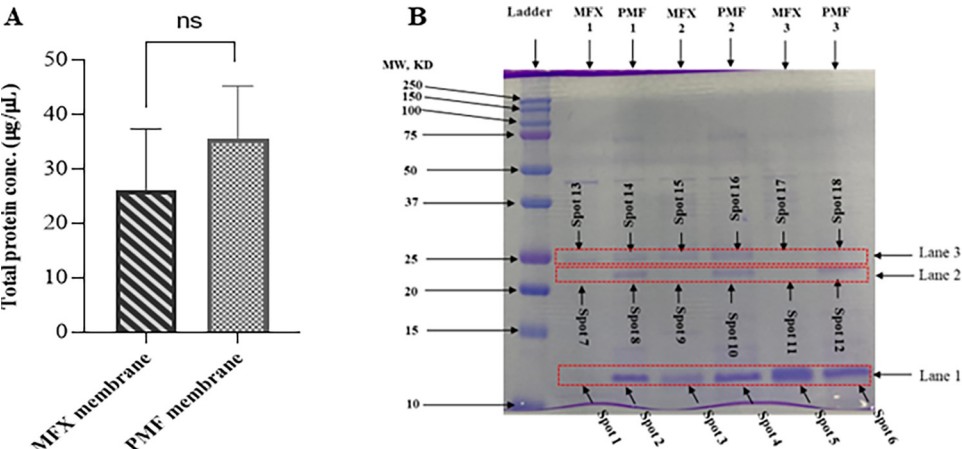

**Fig 1. Total protein concentration and SDS-PAGE of PMMA & PES.** (A) The graph shows the total protein concentration levels in dialyzer membranes. Samples represented as the mean (n = 3) with ±SD, *P< 0.05, PES compared with PMMA by t-test, ns = Not significant. (B) Individual samples of patients in reducing buffer solution from each PES and PMMA membrane were analyzed independently by SDS-PAGE and stained with Coomassie blue. The gel shows the range of protein removal of the membranes between 10–25 kDa. Detected proteins are indicated with numbers (Lane 1–3 and spot no. 1–12) and were processed for identification by LC-MS/MS. The numbers on the left represent the apparent molecular mass in kDa.

mM ammonium bicarbonate) and 200 μL acetonitrile: ultrapure water (1:1, v/v), respectively, and the gel piece was dried in a centrifugal concentrator. Then, 50 μl of trypsin solution [Protein to enzyme ratio 20:1 (w/w), (Promega, Madison, WA, USA)] was added and left on ice for 30 min to allow the solution to penetrate the gel, followed by discarding the standard solution and keeping the gel. Then, 45 μl extraction buffer (50 mM ammonium bicarbonate) was added and incubated at 37˚C overnight (12h) for gel digestion. The next day, the buffer solution was transferred to a tube. Then, 50 μl of extraction buffer (acetonitrile: trifluoroacetic acid: ultrapure water, 10:1:9, v/v/v) was added to the gel and shaken at room temperature for 30 min. The extraction buffer was mixed with the transferred buffer solution (collected previously). The same procedure was repeated with 50 μl of extraction buffer. The protein digests were desalted using the MonoSpin C18 column (GL Sciences, Tokyo, Japan), and the eluted acetonitrile solution was dried in a centrifugal concentrator. Then, dried peptides were resuspended in 100 μl of 0.1% formic acid, and the sample was collected for LC-MS/MS analysis.

## LC-MS/MS analysis

The samples were analyzed by online nanoflow liquid chromatography-tandem mass spectrometry utilizing a Q-Exactive mass spectrometer (Thermo-Fisher Scientific) coupled online to an Easy-nLC 1200 (Thermo Fisher Scientific). The mobile phases contain 0.1% formic acid in 100% $H_2O$ (solvent A) and 0.1% formic acid in 80% acetonitrile (solvent B). The 18 μl sample was loaded onto an Acclaim Pep Map 100 C18 trap column (3μm x 0.075mm x 20 mm; Thermo Fisher Scientific) and separated by a 12.5 cm capillary column NTCC-360/75-3-125 (C18, 3 um, 0.075 mm x 125 mm; Nikkyo Technos) at a constant flow rate of 300 nL/min with a gradient of solvent B changing from 0 to 35% for 50 min, from 35–100% for the next 5 minutes and the column was then washed with 100% solvent B for 10 minutes. LC eluent was sprayed into the MS instrument with a glass emitter tip (PicoTip, New Objective). ESI source settings were a spray voltage of 4.3 kV (positive-ion mode) and a capillary temperature of 320˚C. The Q Exactive instrument was operated to automatically switch between full MS and data-dependent MS/MS scan. Complete scan data were acquired at a resolution of 70,000 from

350 to 1800 *m/z*. The full scan's automatic gain control (AGC) target was set to 3e6. The MS/MS data acquisition parameters were set as follows: AGC target value, le5; scan range, 200 to 2000 m/z; resolution, 17,500 m/z; fixed injection time, 55ms; isolation window, 1.6 *m/z*. Dynamic exclusion time for a data-dependent scan was selected 20s, and charge exclusion was set to unassigned, 1, 5–8, >8.

### Protein identification and analysis

Raw MS/MS data files were analyzed using the default parameters of Proteome Discoverer 2.2 (Thermo Scientific). Mascot search engine (Matrix Science, London, UK) was employed to identify proteins. Search parameters were defined as follows: database, Homo sapiens (human); protein database (updated on Mar 10, 2019); precursor mass tolerance, ten ppm; mass tolerance for MS/MS, 0.02 Da; trypsin digestion with one missed cleavage allowed; carbamidomethylation of cysteine. The percolator was used for peptide validation based on the q-value. For protein identification, high-confidence peptides were employed, and a target false discovery rate (FDR) threshold of 1% was selected at the peptide level. Proteins with at least two unique peptides were used for protein identification. Mass spectra were analyzed with Thermo Xcalibur™ software.

### Western blot analysis of IL-6

Protein samples (20 μg/well) of all used hemodiafilters (patients 1 to 3) and precision plus protein dual color ladders were loaded into 12% SDS-PAGE gels. Then, the proteins were transferred to the polyvinylidene difluoride membrane (Cytiva, Tokyo, Japan). Membranes were blocked using blocking buffer (1X TBS, pH 7.4, 0.1% v/v Tween-20, with 0.5% w/v skim milk) with continuous shaking for 60 min at room temperature and then incubated with shaking overnight at four ˚C using the IL-6 polyclonal antibody (1:1000 dilution). Removal of the excess primary antibody was carried out by washing the membranes in TBST three times for 15 min each. The secondary antibody (horseradish peroxidase-conjugated anti-rabbit antibody) diluted 1:5000 was incubated with the Membrane in TBST with shaking for 60 min at RT. The excess secondary antibodies were cleaned by washing the membranes in TBST three times for 15 min each. Membranes were exposed to chemiluminescence (ECL) reagent (TK274306, Thermo Scientific) for 1 min at room temperature. Subsequently, they detected the bands by the FUSION FX imaging system (Vilber Lourmat, Collégien, Seine-et-Marne, France). Analysis of the area value of each band was conducted using ImageJ software.

### Statistical analysis

Measurement data were analyzed using GraphPad Prism 8.4.2 (GraphPad Software, LaJolla, CA, USA) statistical software.

## Result

### Protein concentration of hemodiafilter extracts

We applied the Pierce BCA assay to measure the total protein concentration in the two types of dialyzer membrane extracts. The adsorbed protein concentrations in the PMMA and PES membranes were 35.6±7.9 μg/μL and 26.1±9.2 μg/μL, with a correlation coefficient of 0.992 (Figs 1A and S1).

### Isolation of protein by SDS-PAGE

SDS–PAGE scanning revealed three major protein peaks between the 10–15 and 20–25 kDa MW. We observed distinct variations in total protein adsorption between both dialyzers,

**Table 3. Identification of candidate proteins of PMMA and PES membrane in LC-MS/MS analysis.**

| SL No | Protein name | Spot number | Protein accession no. | Calc. pI | PSMs | MW (kDa) | Mascot score | Sequence Coverage (%) |
|---|---|---|---|---|---|---|---|---|
| 1 | β2-MG | Lane 1 (Spot 1–6) | P61769 | 6.52 | 11 | 13.7 | 219 | 17 |
| 2 | Dermcidin OS | Lane 1(Spot 1–6) | P81605 | 6.54 | 6 | 11.3 | 130 | 17 |
| 3 | Amyloid A-1 | Lane 1(Spot 1–6) | P0DJI8 | 6.79 | 14 | 13.6 | 326 | 25 |
| 4 | RBP-4 | Lane 2(Spot 7–12) | P02753 | 6.07 | 80 | 23 | 1718 | 24 |
| 5 | Immunoglobulin lambda-1 light chain | Lane 3(Spot 13–18) | P0DOX8 | 6.76 | 24 | 22.8 | 421 | 12 |

In lane 1 (Spot no. 1–6), a mixture of three proteins β2-MG, Dermcidin, and Amyloid A-1 was identified. In lane 2(Spot no. 7–12), RBP-4 and in lane 3 (Spot no. 13–18), Ig Lambda-1 light chain. The accession number is from the Swiss-Prot database.

especially in the range of 20–25 kDa MW. Protein adsorption was detected in Lane 1 and Lane 3 on both membranes. However, in Lane 2, protein adsorption was only observed in the PMMA membrane, not in the PES membrane. This finding led us to evaluate the dialyzer's abilities to eliminate substances with MW between 10 and 25 kDa (Fig 1B and S1 Raw images).

### Identification of candidate proteins (Lane 1, 2, and 3)

We determined the plasma protein signature of patients treated with PMMA and PES membranes using gel digestion and LC-MS/MS analysis. In Lane 1, we identified a mixture of three proteins: β2 microglobulin (β2-MG), dermcidin, and amyloid A-1. The PMMA membrane exhibited significantly higher adsorption of β2-MG and dermcidin, while the PES membrane disclosed greater adsorption of amyloid A-1(Table 3, Figs 2A, 2B, 2C; S2, S3 and S4). In Lane 2, RBP-4 was only adsorbed on the PMMA membrane, while not the PES membrane (Table 3, Figs 2D and S5). In Lane 3, we observed a significantly higher abundance of immunoglobulin λ-1 light chain on the PMMA membrane than on the PES membrane (Table 3, Figs 2E and S6).

### Detection of IL-6

We applied the western blot technique to detect IL-6 Protein in each extract and found an obvious difference between the two hemodiafilters (Fig 3A and S1 Raw images). In patients 1 and 2, the PMMA membrane showed approximately 6- and 5-fold higher amounts of IL-6 than the PES membrane. Furthermore, in patient 3, whose serum C-reactive Protein was elevated to 15.67 mg/dl, IL-6 protein was detected about a 118-fold higher amount compared to the PES membrane (Fig 3B, S1 Table).

### Discussion

In the PMMA membrane, protein adsorption is caused by the occlusion of protein molecules into pores of the homogeneous membrane structure. PMMA filters can also adsorb substances in a broader molecular weight range than PS membrane substances [21]. In this study, we confirmed that the PMMA membrane can adsorb circulating proteins directly ranging from 10 to 25 kDa MW when compared to the PES membrane by nano-HPLC MS/MS (Fig 1).

LC-MS/MS is useful to compare the changes in serum protein profiles between low-flux and high-flux HD [22]. LC-MS/MS can assess the efficacy of protein removal from dialyzer membranes by analyzing toxic substances [23, 24] and unintentionally lost proteins [25]. In this study, by using mass spectrometric analysis, we first showed that the PMMA membrane exhibited significantly higher adsorption of β2-MG, dermcidin, and immunoglobulin λ-1 light chain. Furthermore, RBP-4 was detectable only in the PMMA membrane (Table 3, Fig 2E).

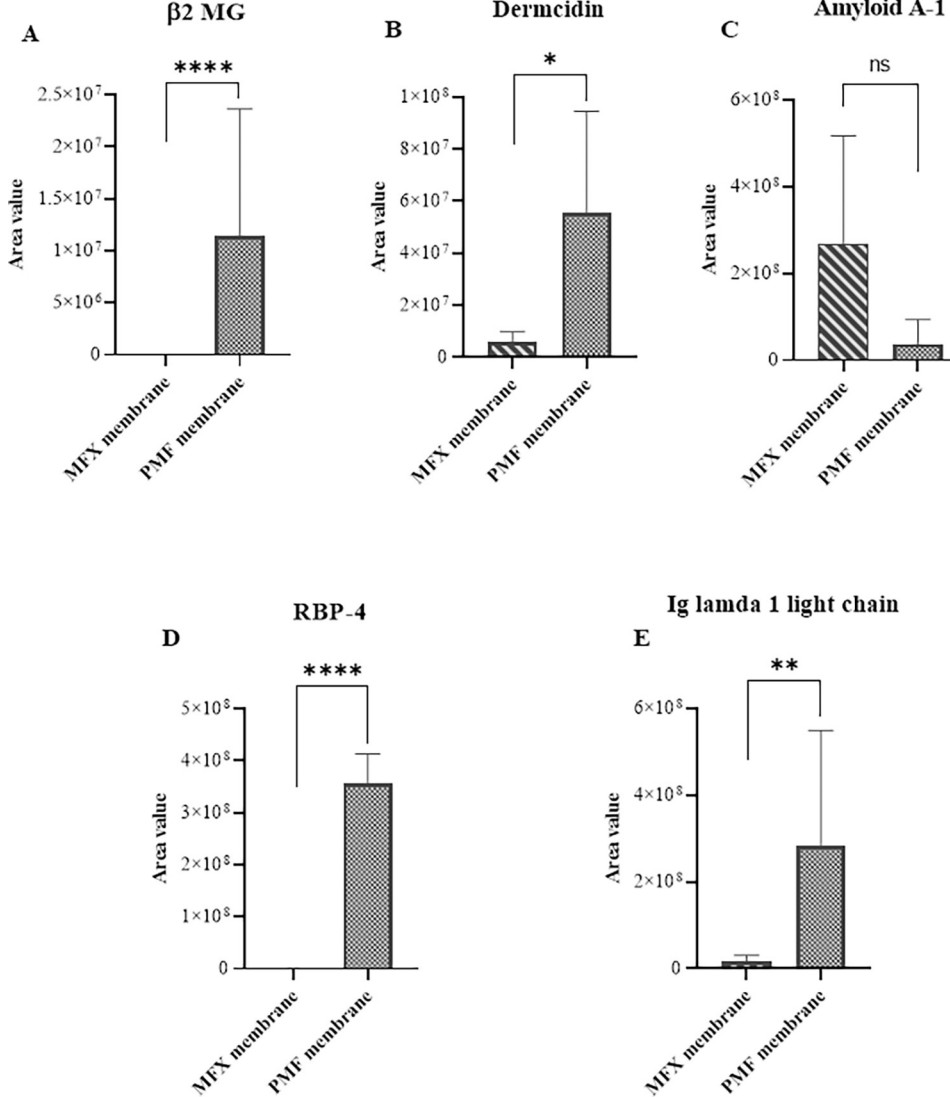

**Fig 2. Comparison of Identified candidate proteins of PMMA and PES membrane.** PMMA membranes show significantly higher adsorption of (A) β2 MG and (B) Dermcidin than the PES membrane, where β2 MG only adsorbs in PMMA. (C) PES membranes adsorb more Amyloid A-1 than PMMA membranes. (D) Only PMMA membranes show the adsorption of RBP4 protein. (E) PMMA membranes show significantly higher adsorption of Immunoglobulin (Ig) lamba-1 light chain protein. Quantification of proteins was performed by Xcaliber ™ software using peak area values for each protein (at the same RT) and PMMA compared with PES by F-test. Error bars represent ±SE; *P< 0.05, **P< 0.01, ***P< 0.001, ****P< 0.0001, ns = not significant.

β2-MG is the key molecule in the development of dialysis-related amyloidosis. An *in vitro* experimental circuit with mini-dialyzers [26] demonstrated that native or glycated β2-MG binding to the PMMA membrane accounted for more than 90% of the removal of native and 80% of glycated β2-MG. In contrast, PS membrane contributed to 55.1 and 57.7%, respectively. In this study, adsorbed β-2MG protein was almost absent in the PES hollow fiber fragment. Thus, β-2MG clearance was mainly mediated through the PMMA filter's adsorption while through the PES filter's highly porous ability.

The adipocytes produce Retinol Binding Protein 4 (RBP4). It is secreted in the circulation, where it binds transthyretin, giving origin to an 80 kDa complex that carries vitamin A

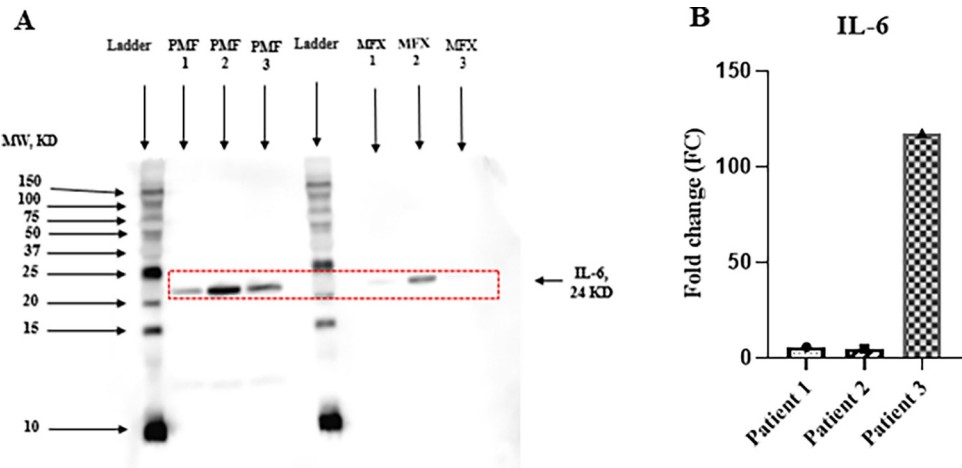

**Fig 3.** (A) Detection of adsorbed IL-6 on the PMMA & PES dialyzer membrane protein extracts by Western blot analysis. IL-6 was detected at the 24 kDa band. The numbers on the left represent the apparent molecular mass in kDa. (B) Quantitative comparison of IL-6 in terms of fold change (FC) of three patients using ImageJ software.

(retinol) to the target tissues. Serum RBP-4 level is elevated in dialysis patients [27, 28]. Elevated RBP-4 is an independent risk factor for arteriovenous fistula (AVF) dysfunction [27], suggesting that RBP-4 may play an essential role in the pathogenesis of neointimal hyperplasia of vascular access. RBP-4 is also associated with insulin resistance. Since HDF treatment granted 20-fold greater intradialytic RBP-4 removal compared to standard HD [28], further studies will be needed to prove RBP-4 removal by predilution online HDF with PMMA membrane may modulate AVF survival time and insulin resistance in ESKD patients.

Serum dermcidin, an antibiotic peptide secreted by sweat glands, is elevated in patients with early-stage melanoma with respect to healthy subjects [29]. Analysis of exhaled breath condensate with LC-MS/MS revealed that dermcidin expression was statistically higher in lung cancer versus healthy volunteers, implying that this non-invasive sampling may serve as biomarkers for lung cancer diagnosis or prognosis [30]. However, the role of elevated serum dermcidin is unclear in ESKD patients.

In general, circulating blood and filtrates are drawn from the extracorporeal circuit at the inlet and outlet of the hemofilter after initiating dialysis to estimate cytokine clearance of hemofilter adsorption [31]. However, the adsorption capacity of cytokine in the circuit model may change over time, thereby leading to an inaccurate clearance. A recent in vitro study disclosed that PMMA hollow fiber can directly adsorb IL-6 about 4-fold higher than PS hollow fiber [18]. So, in this study, we first compared adsorbed proteins on hollow fiber fragments between PMMA and PES collected just after the end of the OL-HDF session in the same patients. We found that the PMMA membrane exhibited significantly higher adsorption of IL-6 proteins (MW: 24 kDa) compared to the PES membrane. Especially in patient 3 with marked elevation of serum CRP, the adsorbed amount of IL-6 was almost 118-fold higher than that in the PES membrane (S1 Table, Fig 3), indicating that IL-6 absorption capacity was upregulated when cytokine overproduction occurred.

Serum amyloid A protein is the most prominent acute phase reactant. In this study, the PES membrane showed higher adsorption of amyloid A-1 protein than other proteins. The mechanism because the PES membrane adsorbed amyloid A-1 protein was unclear. Because the PES membrane does not have a strong negative charge electrically, the pore size and the chemical structure of the PES membrane may be related to its adsorption ability.

There are some limitations to this study. Initially, we conducted this study using a small number of patients. Secondly, we could not assess a relationship between IL-6 adsorption and clinical symptoms between the two membranes due to a short-term cross-over study. Finally, we did not test other pro-inflammatory cytokines.

## Conclusion

Using the LC-MS/MS technique, we showed that PMMA membrane can directly adsorb circulating proteins weighed from 10 to 25 kDa. In particular, IL-6 Protein was absorbed, to a greater extent, by PMMA than by the PES membrane. These findings suggest that PMMA-based OL-HDF therapy may be useful to control inflammation in ESKD patients.

## Supporting information

**S1 Table. Related to Fig 3B.** Comparative detection of IL-6 is highly variable between the MFX and PMF membranes. The area value of each band of IL-6 on the MFX & PMF membrane was analyzed using ImageJ software. Here, FC = Fold change.
(TIF)

**S1 Fig. Related to the Fig 1A.** Calibration curve of BSA standard to determine the total protein concentration.
(TIF)

**S2 Fig. Related to Fig 2A and Table 3.** LC-MS/MS analysis of matched protein β2 MG. (**A**) Representative extract ion chromatogram (EIC) is shown hemodiafilter-bias ion m/z **374.81– 375.96.** Red squares show the ranges of retention time (RT) of targeted ions. (**B**) Mass spectra of target ions. (C) MS/MS spectra of target monoisotopic ions. (D) "VNHVTLSQPK" is identified as a hemodiafilter-specific peptide.
(TIF)

**S3 Fig. Related to Fig 2B and Table 3.** LC-MS/MS analysis of matched protein Dermcidin. (A) Representative extract ion chromatogram (EIC) is shown hemodiafilter-bias ion m/z 564.62–566.49. Red squares show the ranges of retention time (RT) of targeted ions. (B) Mass spectra of target ions. (C) MS/MS spectra of target monoisotopic ions. (D) "ENAGEDPGLAR" is identified as a hemodiafilter-specific peptide.
(TIF)

**S4 Fig. Related to Fig 2C and Table 3.** LC-MS/MS analysis of matched protein Serum amyloid A-1. (A) Representative extract ion chromatogram (EIC) is shown hemodiafilter-bias ion m/z 498.69–500.31. Red squares show the ranges of retention time (RT) of targeted ions. (B) Mass spectra of target ions. (C) MS/MS spectra of target monoisotopic ions. (D) "ENAGEDP-GLAR" is identified as a hemodiafilter-specific peptide.
(TIF)

**S5 Fig. Related to Fig 2D and Table 3.** LC-MS/MS analysis of matched protein Retinol binding protein-4. (A) Representative extract ion chromatogram (EIC) is shown hemodiafilter-bias ion m/z 583.1–583.4. Red squares show the ranges of retention time (RT) of targeted ions. (**B**) Mass spectra of target ions. (C) MS/MS spectra of target monoisotopic ions. (D) "DPNGLPPEAQK" is identified as a hemodiafilter-specific peptide.
(TIF)

**S6 Fig. Related to Fig 2E and Table 3.** LC-MS/MS analysis of matched protein Immunoglobin lambda-1 light chain. (A) Representative extract ion chromatogram (EIC) is shown

hemodiafilter-bias ion m/z 495.25–496.26. Red squares show the ranges of retention time (RT) of targeted ions. (B) Mass spectra of target ions. (C) MS/MS spectra of target monoisotopic ions. (D) "AGVETTTPSK" is identified as a hemodiafilter-specific peptide.
(TIF)

**S7 Fig. Related to Fig 3.** Matched protein Cytokine-like protein from the Excel sheet of Proteome Discover 2.2.
(TIF)

**S1 Raw images. Related to Figs 1B and 3A.** SDS-PAGE of membrane extracts (Raw Fig 1) and Western blot analysis of adsorbed IL-6 on the dialyzer membranes (Raw Fig 2).
(PDF)

## Acknowledgments

We thank all the members of Mitsutoshi Setou's lab for their support and help. We thank Advanced Research Facilities & Services (ARFS) members, Hamamatsu University School of Medicine for technical assistance. Also, we express our gratitude to Mr. Suzuki, the medical engineer, and the supporting personnel at the Blood Purification Therapy Department, Hamamatsu University School of Medicine.

## Author Contributions

**Conceptualization:** Akihiko Kato, Tomohito Sato, Mitsutoshi Setou.

**Data curation:** Md. Shoriful Islam, Shingo Ema, Md. Mahamodun Nabi.

**Formal analysis:** Md. Shoriful Islam, Md. Mahamodun Nabi, Md. Muedur Rahman.

**Funding acquisition:** Akihiko Kato, Tomohito Sato, Mitsutoshi Setou.

**Investigation:** Md. Shoriful Islam, Shingo Ema, Md. Mahamodun Nabi, Md. Muedur Rahman, A. S. M. Waliullah, Jing Yan, Takumi Sakamoto, Yutaka Takahashi, Tomohito Sato, Mitsutoshi Setou.

**Methodology:** Md. Shoriful Islam, Tomohito Sato, Tomoaki Kahyo, Mitsutoshi Setou.

**Project administration:** Akihiko Kato, Tomohito Sato, Mitsutoshi Setou.

**Resources:** Shingo Ema, Yutaka Takahashi, Akihiko Kato, Mitsutoshi Setou.

**Supervision:** Yutaka Takahashi, Tomohito Sato, Tomoaki Kahyo, Mitsutoshi Setou.

**Writing – original draft:** Md. Shoriful Islam, Md. Mahamodun Nabi, Rafia Ferdous, Tomohito Sato.

**Writing – review & editing:** Md. Shoriful Islam, Shingo Ema, Md. Mahamodun Nabi, Md. Muedur Rahman, A. S. M. Waliullah, Jing Yan, Rafia Ferdous, Takumi Sakamoto, Tomohito Sato, Tomoaki Kahyo.

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
