## [Decision Letter · Decision Letter 0]

23 Apr 2024

PONE-D-24-05407Comparative analyses of adsorbed circulating proteins in the PMMA and PES hemodiafilters in patients on predilution online hemodiafiltrationPLOS ONE

Dear Dr. Setou,

Thank you for submitting your manuscript to PLOS ONE. After careful consideration, we feel that it has merit but does not fully meet PLOS ONE’s publication criteria as it currently stands. Therefore, we invite you to submit a revised version of the manuscript that addresses the points raised during the review process.

We look forward to receiving your revised manuscript.

Kind regards,

Robert Chapman, Ph.D.

Academic Editor

PLOS ONE

Journal Requirements:

This work was supported by the MEXT Project to promote public utilization of advanced research infrastructure (Imaging Platform) (Grant Number JPMXS0410300220) and by the AMED project (Grant Number 21ak0101179), Japan. The funders had no role in the study design, data collection and analysis, preparation of the manuscript, or decision to publish. 

Additional Editor Comments:

This manuscript is well written, and should meet the publication criteria with a few minor edits in response to the two reviewers

Reviewers' comments:

Reviewer's Responses to Questions

**Comments to the Author**

1. Is the manuscript technically sound, and do the data support the conclusions?

Reviewer #1: Partly

Reviewer #2: Yes

2. Has the statistical analysis been performed appropriately and rigorously? 

Reviewer #1: No

Reviewer #2: Yes

3. Have the authors made all data underlying the findings in their manuscript fully available?

Reviewer #1: Yes

Reviewer #2: Yes

4. Is the manuscript presented in an intelligible fashion and written in standard English?

Reviewer #1: Yes

Reviewer #2: Yes

5. Review Comments to the Author

**Reviewer #1:** The authors clarified, in patients undergoing online HDF (Hemodiafiltration), what substances are adsorbed by PES (Polyethersulfone) and PMMA (Polymethylmethacrylate) membranes respectively, using mass spectrometry. The method of identifying substances adsorbed on the dialysis membrane is considered to be without question and valid. However, in dialysis, the removal of specific substances is significantly influenced not only by adsorption to the dialysis membrane but also by excretion into the filtrate. In this study, the discussion focuses solely on adsorption, and it is deemed meaningless to argue the clinical significance based on the difference in the amount of substances adsorbed.

**Reviewer #2:** This manuscript by Shoriful Islam et al. provides a succinct study of proteins absorbed by dialysis membranes in the treatment of patients with kidney disease. The authors assess three patients using two different membrane materials, and analyse the absorbed proteins using standard SDS-PAGE and LC-MS/MS proteomics. They were able to find differences in protein absorption between the two membrane materials.

The experimental procedure has been well written with all the relevant details for the proteomics analysis that will allow reproducibility. MS/MS traces have been included in the supporting information, and details of the Mascot searches are also included.

I do have some questions however, before I can recommend publication.

- What was the rationale for first placing the patients with the PES membrane and then switching to the PMMA membrane, would that not skew the results for the PMMA membrane as the PES might absorb proteins first?

- There are some instances where a PS membrane is discussed, is this a typo for PES, or a different type of membrane?

- Figure 2 quantifies the levels of proteins absorbed onto the different membranes. How was this quantification performed? I couldn't find these details in the experimental or in the results section. Also, was any quantitative analysis of band intensity in the gels performed, or was any discussion on band intensity purely qualitative? I would suggest densitometry analysis through ImageJ for example would be more accurate here.

- The second sentence of the abstract is vague for those not in the field - is the hollow fiber in reference to the dialysis membrane?

6. PLOS authors have the option to publish the peer review history of their article (what does this mean?). If published, this will include your full peer review and any attached files.

Reviewer #1: **Yes: **Noriaki Iino

Reviewer #2: No

---

## [Author Response · Author response to Decision Letter 0]

11 May 2024

Response to Reviewer #1

Comment: The authors clarified, in patients undergoing online HDF (Hemodiafiltration), what substances are adsorbed by PES (Polyethersulfone) and PMMA (Polymethylmethacrylate) membranes respectively, using mass spectrometry. The method of identifying substances adsorbed on the dialysis membrane is considered to be without question and valid. However, in dialysis, the removal of specific substances is significantly influenced not only by adsorption to the dialysis membrane but also by excretion into the filtrate. In this study, the discussion focuses solely on adsorption, and it is deemed meaningless to argue the clinical significance based on the difference in the amount of substances adsorbed.

Author response to comment: 

Thank you for your comment and suggestion. We agree that the removal of specific substances in dialysis is significantly influenced by adsorption to the dialysis membrane and excretion into the filtrate. In this study, since we completely washed out the filtered dialysate filled in the column (Page 7, line 113), the influence of excreted dialysate on the analyses was limited. Therefore, we solely focus on adsorption, and as a future aspect, we are currently working on the excretion into the filtrate. 

Response to Reviewer #2

Comments: This manuscript by Shoriful Islam et al. provides a succinct study of proteins absorbed by dialysis membranes in the treatment of patients with kidney disease. The authors assess three patients using two different membrane materials and analyze the absorbed proteins using standard SDS-PAGE and LC-MS/MS proteomics. They were able to find differences in protein absorption between the two membrane materials.

The experimental procedure has been well written with all the relevant details for the proteomics analysis that will allow reproducibility. MS/MS traces have been included in the supporting information, and details of the Mascot searches are also included.

I do have some questions, however, before I can recommend publication.

Comment #1: What was the rationale for first placing the patients with the PES membrane and then switching to the PMMA membrane, would that not skew the results for the PMMA membrane as the PES might absorb proteins first?

Author response to comment #1: 

Thank you for your comment. All patients had equally undergone OL-HDF using either the PES or PMMA filters at about 2-day intervals on a distinct mid-week day. In addition, both hemodiafilters were brand-new without any re-use (Page 6, line 111). So, we believe there is little possibility to influence the results by our switching protocol.

Comment #2: There are some instances where a PS membrane is discussed, is this a typo for PES, or a different type of membrane? 

Author response to comment #2: 

Thank you for your comment, yes this is a typo for PES (Polysulfones include PS and PES microfiltration membranes having almost similar properties). We corrected our manuscript this time (Page no. 3, Line no. 54). 

Comment #3: Figure 2 quantifies the levels of proteins absorbed onto the different membranes. How was this quantification performed? I couldn't find these details in the experimental or in the results section. Also, was any quantitative analysis of band intensity in the gels performed, or was any discussion on band intensity purely qualitative? I would suggest densitometry analysis through ImageJ for example would be more accurate here. 

Author response to comment #3: 

Thank you for your nice comment and suggestion. In Figure 2, we semi-quantitative/qualitatively analyzed the levels of proteins absorbed onto the different membranes by LCMS. We extracted peak area values for each protein (at the same RT) in both PMMA and PES membranes using Xcaliber TM software (Thermo-fisher Scientific) and quantified the levels of proteins absorbed onto the different membranes. Then, we used peak area values to compare the proteins adsorbed by PMMA and PES membranes (Page no. 11, Line no. 233-4). 

Comment #4: The second sentence of the abstract is vague for those not in the field - is the hollow fiber in reference to the dialysis membrane?

Author response to comment #4: 

Thank you for your comment. We added the explanation that, the hollow fiber is in reference to the dialysis membrane (Page no. 2, Line no. 27).

---

## [Editor Report · Decision Letter 1]

31 May 2024

Comparative analyses of adsorbed circulating proteins in the PMMA and PES hemodiafilters in patients on predilution online hemodiafiltration

PONE-D-24-05407R1

Dear Dr. Setou,

Thank you for your detailed response to the reviewer comments. We’re pleased to inform you that your manuscript has been judged scientifically suitable for publication and will be formally accepted for publication once it meets all outstanding technical requirements.

Kind regards,

Robert Chapman, Ph.D.

Academic Editor

PLOS ONE

---

## [Editor Report · Acceptance letter]

10 Jul 2024

PONE-D-24-05407R1 

PLOS ONE

Dear Dr. Setou, 

I'm pleased to inform you that your manuscript has been deemed suitable for publication in PLOS ONE. Congratulations! Your manuscript is now being handed over to our production team.

Kind regards, 

on behalf of

Dr. Robert Chapman 

Academic Editor

PLOS ONE